# DUAL-STREAM ADAPTERS FOR ANOMALY SEGMENTATION

## ABSTRACT

Anomaly segmentation aims to identify pixels of objects not present during the
model's training. Recent approaches address this task using mask-based archi-
tectures, but these methods have high training costs due to the large transformer
backbones involved. While vision adapters can help reduce training costs, they
are not specialized for this task, leading to inferior performance. In this work,
we propose Dual-Stream Adapters (DSA), a vision adapter tailored for anomaly
segmentation. DSA extracts both in-distribution and out-of-distribution features
via (i) an anomaly prior module that produces separate initial embeddings for the
two streams; and (ii) a dual-stream feature refinement that implicitly guides the
separation of in-distribution from out-of-distribution features. We train DSA us-
ing a novel hyperbolic loss function that provides supervised guidance for differ-
entiating in-distribution and out-of-distribution features. Experiments on various
benchmarks show that dual-stream adapters achieve the best results while reduc-
ing training parameters by 38% w.r.t. the previous state-of-the-art.

## 1 INTRODUCTION

Anomaly segmentation of driving scenes is a task that aims at segmenting, with pixel-level granular-
ity, objects in road scenes not belonging to any category from the training distribution (Ackermann
et al., 2023; Blum et al., 2021; Gal & Ghahramani, 2016; Grcić et al., 2022). This is crucial because,
given an object category that is unseen during training (e.g., a cow), a standard semantic segmenta-
tion model would erroneously classify it as one of its known classes (e.g., car, road), being unable
to flag a potential danger. On the other hand, an anomaly segmentation model should segment such
unseen objects, identifying them as anomalies. Although this task requires making predictions at
the pixel level (anomalous pixel / not anomalous pixel), few recent methods have demonstrated that
it is advantageous to resort to mask-architectures to directly identify anomalies as whole masks (Rai
et al., 2023a;b; Ackermann et al., 2023; Nayal et al., 2023). However, the improvements achieved by
these architectures largely depend on training large visual transformer backbones (Liu et al., 2021;
Dosovitskiy et al., 2021) with a significant number of training parameters and, consequently, a sub-
stantial training cost (cf. Fig. 1). Thus, we aim to develop an anomaly segmentation method that is
more efficient at training time (with fewer trainable parameters).

To achieve this goal, we take inspiration from vision transformer adapters (Chen et al., 2023),
which have been shown to be effective for dense prediction tasks, while also being efficient in
terms of training parameters. However, these adapters are meant for general computer vision tasks
and are not specialized for anomaly segmentation, a task that not only requires dense predictions,
but it also needs to model the concepts of in-distribution and out-of-distribution. Therefore, in
this work, we present the *Dual-Stream Adapter*, a novel adapter tailored for anomaly segmenta-
tion. Its architecture consists of two key components: i) an *anomaly prior module* that initially
learns in-distribution and out-of-distribution features through distinct feature-level encodings; ii) a
*dual-stream feature refinement module* that refines these features and combines them with the pow-
erful representations of the frozen ViT backbone. This architecture is complemented by a novel hy-
perbolic loss function that guides the learning process, using void labels as supervision, to increase
uncertainty for OOD pixels, while reducing uncertainty for ID pixels. We perform extensive experi-
ments on standard anomaly segmentation benchmarks (Fishyscapes (Blum et al., 2021), Segment Me
If You Can (Chan et al., 2021a), Road Anomaly Lis et al. (2019)), showing that Dual-Stream Adapter

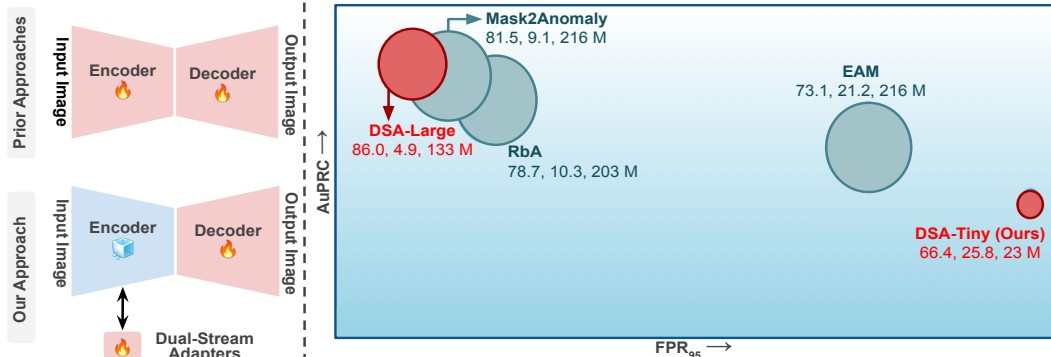

Figure 1: **Left:** State-of-the-methods (Grcić et al., 2023; Rai et al., 2023b; Nayal et al., 2023) requires training of large backbones, such as Swin-L (Liu et al., 2021), with huge number of trainable parameters. We propose a **Dual-Stream Adapter** having a frozen pre-trained backbone to perform anomaly segmentation and thus making the training parameter efficient. **Right:** We observe that our DSA-Large performs best among the recent state-of-the-art methods with 38% lower training parameters. Triplet of numbers shown for each method represents AuPRC, FPR$_{95}$, and number of trainable parameters in millions, respectively. Best method have lowest FPR$_{95}$ and highest AuPRC.

achieves the best results among all anomaly segmentation methods with 38% lower training parameters, when compared to the best baseline method (cf. Fig. 1).

**To summarize, our contribution** is the Dual-Stream Adapter, an adapter-based anomaly segmentation method that achieves state-of-the-art results, but with fewer training parameters than the competitors, while also maintaining higher accuracy on the in-distribution data. It consists of a symmetric architecture that allows to disentangle of in-distribution and out-of-distribution features, and it leverages a novel hyperbolic loss that uses the void/background labels in the training dataset as supervision to separate in-distribution and out-of-distribution features.

## 2 RELATED WORK

**Anomaly Segmentation** is synonymous with the problem of pixel-level anomaly detection. Hendrycks & Gimpel (2017) first delved into this problem by leveraging techniques originally developed for image-wide anomaly detection. Their solution, based on the softmax prediction probability, is still considered a standard baseline for anomaly segmentation. Since then, other techniques have been used to segment anomalies, such as deep ensembles (Fort et al., 2019), Bayesian deep learning (Mukhoti & Gal, 2018; Gal & Ghahramani, 2016; Tu et al., 2024), and class-logits reasoning (Jung et al., 2021; Hendrycks et al., 2021). Another strategy is to rely on image re-synthesis, by looking at the discrepancy of the generated data w.r.t the in-distribution training data (Lis et al., 2019).

While all the methods discussed so far are completely unsupervised, recent methods improve results via outlier exposure (Bevandić et al., 2018; Di Biase et al., 2021; Zhang et al., 2023), i.e., providing supervision to the model by training on mixed images that contain instances of novel categories (*negative crops*) pasted on them. These negative crops are generally sampled from the MS-COCO (Lin et al., 2014) or ADE-20K (Zhou et al., 2017) datasets, which contain objects absent in in-distribution datasets. Alternatively, when appropriate negative data may be unavailable, synthetic negatives may be used effectively, as demonstrated by Grcić et al. (2021) which introduced a normalizing flow-based method. Afterward, Gudovskiy et al. (2023) improved the prior approach with an energy-based normalizing flow method. Concurrently, DaCUP Vojír & Matas (2023) proposed novel embedding bottleneck and image-conditioned distance features to improve anomaly segmentation.

Current state-of-the-art for anomaly segmentation (Nayal et al., 2023; Yu et al., 2023; Grcić et al., 2023; Ackermann et al., 2023; Rai et al., 2023b;a) leverage mask-based architectures (Cheng et al., 2021; 2022) that perform pixel-wise classification as a direct set prediction, outputting a set of binary masks with corresponding category labels. By using these architectures and by aggregating anomaly

scores on the predicted masks, rather than reasoning on individual pixels, RBA (Nayal et al., 2023), EAM (Grcić et al., 2023), Mask2Anomaly (Rai et al., 2023b;a), and Maskomaly (Ackermann et al., 2023) obtain significant improvements in anomaly segmentation w.r.t. previous solutions. However, the state-of-the-art results rely on using large Transformer backbones, such as ViT (Dosovitskiy et al., 2021) and Swin (Liu et al., 2021), which have a significant number of training parameters, thereby incurring in a substantial training cost.

To address this problem, we propose the Dual-Stream Adapter, an adapter tailored for anomaly segmentation, which directly leverages a large ViT backbone, pre-trained on a large dataset to give robust features without retraining. Our Dual-Stream Adapter retains the advantages of previous mask-based anomaly segmentation methods, while being more training efficient. Moreover, at deployment, this solution enables model reuse, since the same frozen pre-trained backbone can be utilized for other downstream tasks, eliminating the need for multiple specialized versions.

**Adapters.** Learning universal representations that can be specialized to multiple tasks and domains is quintessential in deep learning. Among the techniques developed to transfer learned representations, adapters (Rebuffi et al., 2017; 2018) have emerged as an effective and parameter-efficient solution. The idea behind adapters is to get compact parametrizations of models by partitioning their parameters into two parts: a vector of fixed parameters that capture a universal representation, and a smaller vector of learnable parameters that are domain or task-specific. Adapters have recently gained a lot of traction in Natural Language Processing (NLP), where they have been used as a mechanism to adapt the large language models, such as BERT (Devlin et al., 2019), to many downstream tasks, while using a minimal number of trainable parameters (Houlsby et al., 2019). In computer vision, adapters have been utilized for incremental and multi-task/multi-domain learning (Rebuffi et al., 2017; Rosenfeld & Tsotsos, 2018; Rebuffi et al., 2018). More recently, with the success of vision-language models, a plethora of adapters (Zhang et al., 2022; Sung et al., 2022), for the CLIP architecture (Radford et al., 2021), were introduced to address zero-shot/few-shot classification or vision-language tasks. Similarly to NLP, in computer vision the emergence of large pre-trained models (Kirillov et al., 2023; Oquab et al., 2024) is making the utilization of adapters quite compelling. In fact, fine-tuning these models for possibly many downstream tasks is inefficient not only in terms of required training resources, but also in terms of deployed resources, as a device that must perform multiple tasks would require multiple specialized copies of the same model. Moreover, it was also demonstrated that fine-tuning these large vision models could distort the pre-trained features (Kumar et al., 2022) and lead to poor generalization. This has inspired a few works to use a vanilla ViT architecture (Dosovitskiy et al., 2021) and adapt it to various vision tasks (Li et al., 2021; 2022). Along these lines, Chen et al. (2023) have introduced the ViT-Adapter, which integrates vision-specific inductive biases into the plain ViT, reaching comparable performance to recent vision-specific transformer variants on several tasks, but in a parameter-efficient way.

Unlike the general-purpose ViT-Adapter (Chen et al., 2023), which is designed for multiple vision tasks, we introduce an adapter specifically for anomaly segmentation. Since the anomaly segmentation problem requires the classifier to identify 'none-of-the-above' scenarios, our Dual-Stream Adapter is built to reinforce the separation between in-distribution features and complementary features that represent unfamiliar or anomalous objects. To the best of our knowledge, this is the first adapter explicitly tailored for anomaly segmentation.

## 3 PRELIMINARIES

### 3.1 PROBLEM SETTING

Anomaly segmentation aims to identify within an image those pixels containing objects not encountered during the model's training. To formalize the task, let us consider a training dataset $\mathcal{D} = \{(x_i, y_i)\}_{i=1}^{D}$, where $x_i \in \mathcal{X}$ represents an RGB image from the set $\mathcal{X} \subset \mathbb{R}^{3 \times H \times W}$, and $y_i \in \mathcal{Y} \subset \mathbb{N}^{|\mathcal{Z}| \times H \times W}$ is the corresponding ground truth, assigning a semantic label from a predefined set of categories $\mathcal{Z}$ to each pixel. Here, $H$ and $W$ denote the height and width, respectively, of each sample of $\mathcal{X}$ and $\mathcal{Y}$.

Traditionally, anomaly segmentation has been approached as a per-pixel classification task (Grcić et al., 2022; Tian et al., 2022). This involves first learning a function $f_\theta : \mathcal{X} \to \mathbb{R}^{|\mathcal{Z}| \times H \times W}$ that maps

an image to pixel-wise class scores from the pre-defined set $\mathcal{Z}$, and then applying a non-parametric scoring function $s : \mathbb{R}^{|\mathcal{Z}| \times H \times W} \to [0, 1]^{H \times W}$, to convert these class scores into anomaly scores. Recent state-of-the-art approaches (Nayal et al., 2023; Ackermann et al., 2023; Rai et al., 2023b; Grcić et al., 2023; Rai et al., 2023a; Yu et al., 2023) have reconsidered this pipeline, by leveraging the Mask2Former architecture (Cheng et al., 2022). These methods shift from per-pixel classification to mask-based classification. We briefly summarize this reformulation in the next paragraph, as it forms the foundation of our solution.

## 3.2 MASK2FORMER-BASED ANOMALY SEGMENTATION

Mask2Former (Cheng et al., 2022) is a universal architecture that approaches segmentation as a direct set prediction problem. It groups pixels into $N$ segments by predicting $N$ binary masks and their corresponding category labels (where $N$ is a hyper-parameter). The architecture consists of three main components: (i) an *encoder* that extracts the features from the input image, (ii) a *pixel-decoder* that generates high-resolution per-pixel embeddings from the encoder's low-resolution features, and (iii) a *transformer-decoder* that operates on image features to process object queries. For brevity, we do not delve into a detailed description of these components. Interested readers can refer to Cheng et al. (2022) for a comprehensive explanation of the Mask2Former architecture. Additionally, for the sake of compactness and readability, we henceforth refer to the pixel decoder and transformer decoder collectively as the *decoder* (see Fig. 1-left).

Given this architecture, our aim is to learn the parameters $\theta$ of a function $f_\theta$, composed of an encoding function $h_{\theta_e}$ and a decoding function $g_{\theta_d}$ with $\theta = [\theta_e, \theta_d]$, that maps an image to a set of binary masks $M \in \mathbb{R}^{N \times (H \times W)}$, and category labels $C \in \mathbb{R}^{N \times |\mathcal{Z}|}$. Formally, we have

$$f_\theta : \mathcal{X} \to (\mathbb{R}^{N \times (H \times W)}, \ \mathbb{R}^{N \times |\mathcal{Z}|})$$
$$x \mapsto g_{\theta_d} \circ h_{\theta_e}(x) = (M, C) \tag{1}$$

The anomaly scores are finally obtained through a non-parametric scoring function $s : (\mathbb{R}^{N \times (H \times W)}, \ \mathbb{R}^{N \times |\mathcal{Z}|}) \to \mathbb{R}^{H \times W}$ that takes as inputs the binary masks $M$ and class scores $C$. Following Rai et al. (2023b), we implement the scoring function $s$ by applying MSP (Hendrycks & Gimpel, 2017) on the decoder outputs $M$ and $C$, i.e.,

$$s(M, C) = 1 - \max_{\mathcal{Z}} \left( \texttt{softmax}(C)^T \cdot \texttt{sigmoid}(M) \right) \tag{2}$$

Without loss of generality, other scoring functions may be used, e.g., Nayal et al. (2023); Grcić et al. (2023).

This approach has been proven crucial for achieving new state-of-the-art results, especially when using large pre-trained encoders (Nayal et al., 2023; Ackermann et al., 2023; Rai et al., 2023b; Grcić et al., 2023; Rai et al., 2023a; Yu et al., 2023). Despite their flexibility, fine-tuning these large vision encoders is generally a delicate process as it could distort the pre-trained features (Kumar et al., 2022), making them less general and ineffective for other downstream tasks. To alleviate this issue, while maintaining state-of-the-art performance, we propose freezing the pre-trained encoder and inserting an adapter module (Rebuffi et al., 2018; Houlsby et al., 2019) with learnable parameters $\theta_a$, such that $\|\theta_a\| << \|\theta_e\|$ (see Fig. 1-left). Namely, we now seek a map

$$f_\theta : \mathcal{X} \to (\mathbb{R}^{N \times (H \times W)}, \ \mathbb{R}^{N \times |\mathcal{Z}|})$$
$$x \mapsto g_{\theta_d} \circ h_{\bar{\theta}_e, \theta_a}(x) = (M, C) \tag{3}$$

where $\bar{\theta}_e$ denotes the frozen parameters. We model this adapter after the ViT-Adapter (Chen et al., 2023), as explained in the next section.

## 4 DUAL STREAM ADAPTER

The Dual-Stream Adapter (DSA) architecture is based upon the ViT-Adapter structure (Fig. 2). It incorporates an anomaly prior module to capture initial ID and OOD features and dual-stream feature refinement modules that refine these features through interaction with frozen ViT features. We guide the separation of ID and OOD features by training them with a novel hyperbolic loss function. In the following subsections, we discuss these technical novelties in detail. For clarity, we omit the dependency of the adapter modules from the parameters $\theta_a$.

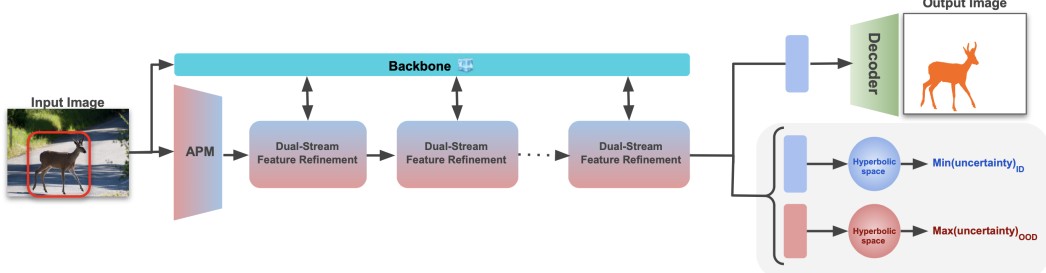

Figure 2: **Dual-Stream Adapter:** Our model consists of a frozen ViT backbone, a decoder, and a dual-stream adapter that extracts two complementary features referring to the in-distribution objects and to the *rest of the objects*, respectively. The dual-stream adapter consists of two components: anomaly prior module (APM) (Sec. 4.1) and a dual-stream feature refinement module (Sec. 4.2).

## 4.1 Anomaly Prior Module

The first part of the DSA is the anomaly prior module (see Fig. 3-left). Our module shares a similar foundation with the ViT-Adapter's Spatial Prior Module, leveraging a ResNet convolutional stem (He et al., 2016) to extract $D$-dimensional feature maps at different spatial resolutions: ($\mathcal{F}_8 \in \mathbb{R}^{D \times \frac{H}{8} \times \frac{W}{8}}$, $\mathcal{F}_{16} \in \mathbb{R}^{D \times \frac{H}{16} \times \frac{W}{16}}$ and $\mathcal{F}_{32} \in \mathbb{R}^{D \times \frac{H}{32} \times \frac{W}{32}}$). The intuition of this design choice is that convolutional operations improve the ability to extract spatial information in vision transformers by capturing local context and providing translation equivariance (Wang et al., 2021). Each of the three feature tensors is further augmented with an additive and learnable *level encoding*. Formally, for a generic feature scale $\mathcal{F}_i$, it is given as $\bar{\mathcal{F}}_i = \mathcal{F}_i + \delta_i$. Where, $\delta_i \in \mathbb{R}^{D \times \frac{H}{i} \times \frac{W}{i}}$ is the corresponding level encoding. All these features are finally flattened and concatenated together.

We diverge from ViT-Adapter by introducing an explicit specialization mechanism. Features are processed in two separate streams with distinct level encodings, enabling them to specialize on either ID or foreign object characteristics (see Fig. 3-left). Formally, for a generic feature scale $\mathcal{F}_i$, we have

$$\bar{\mathcal{F}}_{i,\text{id}} = \mathcal{F}_i + \delta_{i,\text{id}}$$
$$\bar{\mathcal{F}}_{i,\text{ood}} = \mathcal{F}_i + \delta_{i,\text{ood}} \tag{4}$$

where $\delta_{i,\text{id}}, \delta_{i,\text{ood}} \in \mathbb{R}^{D \times \frac{H}{i} \times \frac{W}{i}}$ are the corresponding level encodings, each of them obtained as a distinct $D$-dimensional learnable vector broadcasted along the spatial coordinates. Importantly, the level encoding here acts as a channel-wise bias individually learned for in-distribution and out-of-distribution features, that are added into the features to retain information on their resolution level within the feature tokens.

The feature tensors of the two streams are finally flattened and concatenated separately, yielding two arrays of $D$-dimensional feature tokens, i.e.,

$$\bar{\mathcal{F}}_{8,\text{id}}, \bar{\mathcal{F}}_{16,\text{id}}, \bar{\mathcal{F}}_{32,\text{id}} \mapsto \mathcal{F}^1_{\text{id}} \in \mathbb{R}^{(\frac{HW}{8^2} + \frac{HW}{16^2} + \frac{HW}{32^2}) \times D}$$
$$\bar{\mathcal{F}}_{8,\text{ood}}, \bar{\mathcal{F}}_{16,\text{ood}}, \bar{\mathcal{F}}_{32,\text{ood}} \mapsto \mathcal{F}^1_{\text{ood}} \in \mathbb{R}^{(\frac{HW}{8^2} + \frac{HW}{16^2} + \frac{HW}{32^2}) \times D} \tag{5}$$

With the anomaly prior module, the architecture can learn strong prior knowledge about the ID and OOD features evident from Tab. 5 (a).

## 4.2 Dual Stream Feature Refinement

The original ViT-Adapter (Chen et al., 2023) uses an interaction module consisting of cross-attentions to combine the spatial information from the prior module and the features from the frozen ViT. For our problem, we want to combine the frozen ViT features with the spatial in-distribution (ID) and out-of-distribution (OOD) information, while at the same time disentangling them. Therefore, we design a module based on cross-attention with a symmetric structure to handle the dual streams information.

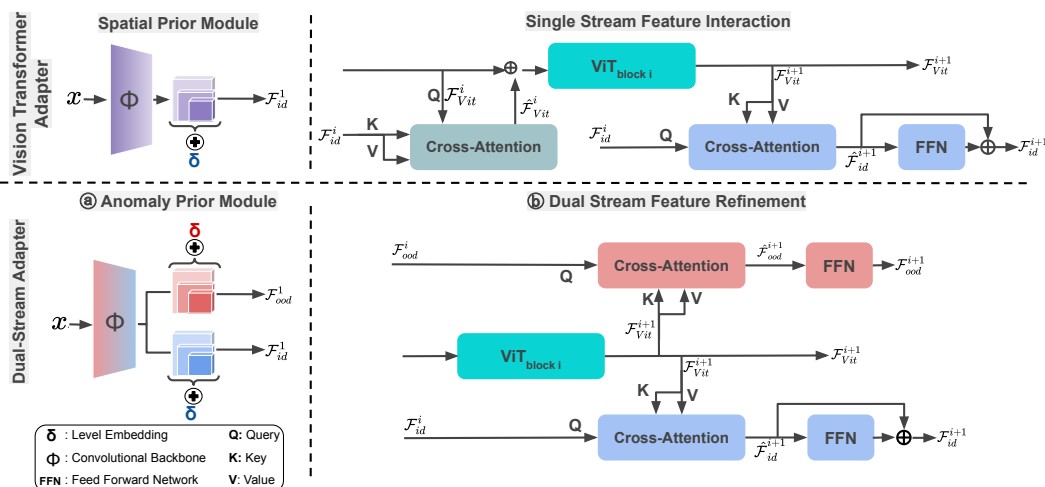

Figure 3: **Above:** Shows the spatial prior module and adapter architecture present in the existing vision adapters architecture (Chen et al., 2023). **Below:** We present novel components of our dual-stream adapter. *(a)Anomaly Prior Module (Sec. 4.1):* learn to extract the initial ID and OOD features from the input image. *(b)Dual-Stream Feature Refinement (Sec. 4.2):* takes in the initial ID and OOD features from the anomaly prior module. The features are refined by passing it through a set of anomaly adapters and augmenting it with the ViT backbone features.

Each of this modules, which we call dual-stream feature refinement, consists of two symmetric streams: one for the ID features and the other for the OOD features. Without loss of generality, let us consider the ID stream for the $i$-th dual-stream feature refinement module. To combine the spatial ID prior with the ViT features, we use a cross-attention layer, where the keys and values are given from features produced by the $(i + 1)$-th ViT block, i.e., $\mathcal{F}_{\text{Vit}}^{i+1}$. The spatial features $\mathcal{F}_{\text{id}}^{i}$ are used as queries to reintegrate relevant multi-scale features from $\mathcal{F}_{\text{Vit}}^{i+1}$ into the tokens of $\mathcal{F}_{\text{id}}^{i+1}$. The same sequence of operations is implemented for the OOD stream. Formally, the whole process is summarized as:

$$\mathcal{F}_{\text{id}}^{i+1} = \hat{\mathcal{F}}_{\text{id}}^{i+1} + \text{FFN}(\hat{\mathcal{F}}_{\text{id}}^{i+1})$$

$$\textit{i.d. extractor} \quad \hat{\mathcal{F}}_{\text{id}}^{i+1} = \mathcal{F}_{\text{id}}^{i} + \text{Attention}(\underbrace{\mathcal{F}_{\text{id}}^{i}}_{Q}, \underbrace{\mathcal{F}_{\text{ViT}}^{i+1}}_{K}, \underbrace{\mathcal{F}_{\text{ViT}}^{i+1}}_{V}) \quad (6)$$

$$\mathcal{F}_{\text{ood}}^{i+1} = \hat{\mathcal{F}}_{\text{ood}}^{i+1} + \text{FFN}(\hat{\mathcal{F}}_{\text{ood}}^{i+1})$$

$$\textit{o.o.d. extractor} \quad \hat{\mathcal{F}}_{\text{ood}}^{i+1} = \mathcal{F}_{\text{ood}}^{i} + \text{Attention}(\underbrace{\mathcal{F}_{\text{ood}}^{i}}_{Q}, \underbrace{\mathcal{F}_{\text{ViT}}^{i+1}}_{K}, \underbrace{\mathcal{F}_{\text{ViT}}^{i+1}}_{V}) \quad (7)$$

where $\text{Attention}$ denotes the cross-attention block and $\text{FFN}$ is a feed-forward network as in the original ViT-Adapter (Chen et al., 2023).

Besides its dual structure, our module differs from ViT-Adapter's interaction module also for its lack of an injector cross-attention (see Fig. 3) module. This design choice is empirically motivated, as we found that the anomaly segmentation results are similar with and without the injector (cf. Tab. 6 ). However, removing the injector cross-attention results in a more streamlined module, with fewer training parameters. In summary, dual-stream feature refinement helps to refine and specialize the features obtained from anomaly prior to improve anomaly segmentation while retaining a good performance on the in-distribution samples.

### 4.3 UNCERTAINTY BASED HYPERBOLIC LOSS

Although the adapter architecture described so far is designed to generate two streams of features, we need a way to guide the learning process so that these two streams capture characteristics of the

in-distribution and out-of-distribution pixels, respectively. To provide this guidance without relying on extra data, we leverage the void/background class, considering those pixels as OOD whereas the labeled objects are ID, and we design a loss in hyperbolic space.

The hyperbolic space formulation was demonstrated to be effective in a plethora of visual tasks (Atigh et al., 2022; Liu et al., 2020) over Euclidean space. Moreover, it was shown that it enables the use of low-dimensional embeddings without compromising representation power (Nickel & Kiela, 2017), which is particularly well-suited for adapters. Most importantly, Atigh et al. (2022) demonstrated that with hyperbolic embeddings in a Poincaré ball the pixel uncertainty can be obtained by taking the $L2$ norm to the origin of the ball itself. We use this property to design a new loss function termed as uncertainty-based hyperbolic loss ($L_{ubhl}$), that maximizes the uncertainty for OOD pixels and minimizes uncertainty for ID pixels. In the following, we discuss $L_{ubhl}$ in detail.

To define this loss, we first take the final ID ($\mathcal{F}_{id}$) and OOD ($\mathcal{F}_{ood}$) features and project them to the Hyperoblic space using an exponential map (Atigh et al., 2022), i.e.,

$$\mathbb{E}_0(\bar{x}) = tanh(\sqrt{c}||\bar{x}||)(\frac{\bar{x}}{\sqrt{c}||\bar{x}||}) \tag{8}$$

Then, we calculate uncertainty by taking $L2$ norm to the origin of the Poincaré ball of projected features. Finally, $L_{ubhl}$ for a feature point $\bar{x}_i$ can be formulated as:

$$L_{ubhl}(\bar{x}_i) = \begin{cases} ||d(\bar{x}_i, 0)||_2 & \text{if } \bar{x}_i \in \mathcal{F}_{ood} \\ -||d(\bar{x}_i, 0)||_2 & \text{otherwise,} \end{cases} \tag{9}$$

$d(\bar{x}_i, 0)$ is the $L2$ norm distance to the origin of the Poincaré ball. Intuitively, this loss pushes the ID features away from the origin of the ball, whereas the OOD features are pulled towards the origin. We add $L_{ubhl}$ to the total training loss (that is the same as Mask2Former (Cheng et al., 2022)) formulated as:

$$L_{total} = \lambda_{ubhl}L_{ubhl} + L_{mask} + \lambda_{class}L_{class} \tag{10}$$

Here, $L_{mask}$ is the mask loss which is a combination of the binary cross entropy loss and dice loss, and $L_{class}$ is the classification loss weighted by $\lambda_{class} = 2$. $\lambda_{ubhl}$ is the weight of $L_{ubhl}$ which is set to 0.1.

| Methods | Road Anomaly | | SMIYC RA-21 | | SMIYC RO-21 | | FS L&F | | FS Static | | Average | |
|---|---|---|---|---|---|---|---|---|---|---|---|---|
| | AuPRC↑ | FPR$_{95}$↓ | AuPRC↑ | FPR$_{95}$↓ | AuPRC↑ | FPR$_{95}$↓ | AuPRC↑ | FPR$_{95}$↓ | AuPRC↑ | FPR$_{95}$↓ | AuPRC↑ | FPR$_{95}$↓ |
| Fine-Tuned | 28.4 | 85.4 | 57.4 | 82.2 | 63.6 | 2.6 | 44.1 | 33.1 | 29.5 | 88.6 | 44.6 | 58.3 |
| Side Adapters (Xu et al., 2023) | 23.1 | 85.2 | **60.3** | **77.3** | 29.3 | 12.1 | **45.1** | 23.6 | 25.2 | 88.3 | 36.6 | 57.3 |
| ViT Adapters (Chen et al., 2023) | 26.0 | 88.3 | 57.1 | 90.7 | **60.3** | 7.5 | 42.5 | 27.0 | 31.5 | 33.1 | 43.4 | 49.3 |
| DSA-Tiny (Ours) | **36.7** | **77.3** | 58.3 | 80.4 | 59.6 | **3.0** | 44.0 | **20.0** | **40.1** | **19.9** | **47.7** | **40.1** |

Table 1: **Quantitative comparison of adapters:** We present the performance comparison of the baseline vision adapters (Side Adapters (Xu et al., 2023), ViT Adapters (Chen et al., 2023)) versus our proposed adapters DSA-Tiny. Best results are in **bold**.

## 5 EXPERIMENTS

**Dataset and Evaluation Metrics:** We train all the models on Cityscapes (Cordts et al., 2016) as an inlier dataset and evaluate its performance on: a) Road Anomaly (Lis et al., 2019), b) Fishyscapes (FS) (Blum et al., 2021) benchmark having FS Lost and Found (FS L&F) and FS Static datasets, and c) Segment Me If You Can (SMIYC) benchmark (Chan et al., 2021a) containing SMIYC Road Anomaly 21 (SMIYC RA-21) and SMIYC Road Obstacle 21 (SMIYC RO-21) datasets. We employ AuPRC and FPR$_{95}$ as evaluation metrics. Please refer to appendix A.2 for further details.

**Implementation Details:** Our implementation details of adapters are mainly derived from Cheng et al. (2022); Chen et al. (2023). We introduced two configurations of adapters: Dual-Stream Adapter Tiny (DSA-Tiny) and Dual-Stream Adapter Large (DSA-Large), consisting of ViT-Tiny and ViT-Large backbones, respectively. Please refer to the appendix A.1 for further details.

| Methods | Road Anomaly | | SMIYC RA-21 | | SMIYC RO-21 | | FS L&F | | FS Static | | Average | |
|---|---|---|---|---|---|---|---|---|---|---|---|---|
| | AuPRC↑ | FPR95↓ | AuPRC↑ | FPR95↓ | AuPRC↑ | FPR95↓ | AuPRC↑ | FPR95↓ | AuPRC↑ | FPR95↓ | AuPRC↑ | FPR95↓ |
| Fine-Tuned | 39.1 | 76.4 | 67.2 | 50.8 | 10.7 | 29.8 | 66.2 | 28.3 | 89.1 | 12.3 | 54.4 | 39.5 |
| Side Adapters (Xu et al., 2023) | 44.8 | **62.1** | 61.1 | 45.9 | 53.8 | 41.6 | 47.2 | 40.9 | **88.9** | 15.9 | 59.1 | 41.2 |
| ViT Adapters (Chen et al., 2023) | 37.5 | 78.5 | 60.7 | **24.9** | **79.5** | 16.0 | 44.9 | 42.6 | 80.6 | 27.3 | 60.6 | 37.8 |
| DSA-Tiny (Ours) | **52.3** | 65.6 | **72.1** | 35.0 | 77.9 | **2.6** | **49.5** | **17.4** | 80.3 | **8.6** | **66.4** | **25.8** |

Table 2: **Quantitative comparison of adapters with outlier exposure:** Shows the performance comparison among the baseline adapter methods and our proposed adapter DSA-Tiny in an outlier supervision setting. Best results are in **bold**.

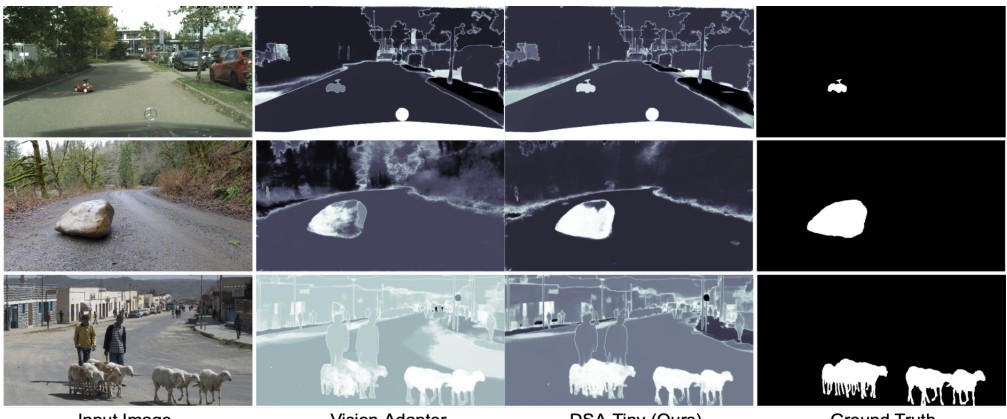

| Input Image | Vision Adapter | DSA-Tiny (Ours) | Ground Truth |

Figure 4: **Qualitative comparison of adapters:** We observe that our proposed DSA-Tiny produces better anomaly segmentation with minimal false positive rates than other vision adapters. Anomalies are represented in white.

| Methods | SMIYC RA-21 | | SMIYC RO-21 | | FS L&F | | FS Static | | Road Anomaly | | Average | |
|---|---|---|---|---|---|---|---|---|---|---|---|---|
| | AuPRC↑ | FPR95↓ | AuPRC↑ | FPR95↓ | AuPRC↑ | FPR95↓ | AuPRC↑ | FPR95↓ | AuPRC↑ | FPR95↓ | AuPRC↑ | FPR95↓ |
| Max Softmax (Hendrycks & Gimpel, 2017) | 27.9 | 72.0 | 15.7 | 16.6 | 4.5 | 40.5 | 19.0 | 23.9 | 15.7 | 71.3 | 16.6 | 44.9 |
| Mahanalobis (Lee et al., 2018) | 20.0 | 86.9 | 20.9 | 13.0 | 56.3 | 11.2 | 27.3 | 11.7 | 14.3 | 81.0 | 27.8 | 40.8 |
| Image Resynthesis (Lis et al., 2019) | 52.2 | 25.9 | 37.7 | 4.7 | 5.7 | 47.7 | 8.0 | 62.7 | - | - | 25.9 | 35.3 |
| MC Dropout (Mukhoti & Gal, 2018) | 28.8 | 69.4 | 4.8 | 50.3 | 12.2 | 32.8 | 42.1 | 13.2 | - | - | 22.0 | 41.4 |
| Learning Embedding (Blum et al., 2021) | 37.5 | 70.7 | 0.8 | 46.3 | 4.1 | 22.3 | 43.5 | 16.8 | - | - | 53.1 | 42.6 |
| SML (Jung et al., 2021) | 46.8 | 39.5 | 3.4 | 36.8 | 36.5 | 14.5 | 48.6 | 16.7 | 17.5 | 70.7 | 22.0 | 41.4 |
| ♠SynBoost (Di Biase et al., 2021) | 56.4 | 61.8 | 81.7 | 4.6 | 40.9 | 34.4 | 48.4 | 47.7 | 38.2 | 64.7 | 21.5 | 39.0 |
| Maximized Entropy (Chan et al., 2021b) | 85.4 | 15.0 | 85.0 | 0.7 | 40.8 | 37.2 | 72.4 | 12.9 | 48.8 | 31.7 | 66.5 | 19.5 |
| ♠JSRNet (Vojir et al., 2021) | 33.6 | 43.8 | 28.0 | 18.5 | 0.2 | 69.3 | 1.4 | 60.4 | **94.4** | **9.2** | 31.5 | 40.2 |
| ♠Dense Hybrid (Grcić et al., 2022) | 77.9 | 9.8 | 87.0 | 0.2 | 63.8 | 6.1 | 60.0 | 4.9 | 31.3 | 63.9 | 64.0 | 17.0 |
| ♠PEBEL (Tian et al., 2022) | 49.1 | 40.8 | 4.9 | 12.6 | 59.8 | 6.4 | 82.7 | 6.8 | 45.1 | 44.5 | 48.3 | 22.2 |
| ♣♠EAM (Grcić et al., 2023) | 76.3 | 93.9 | 66.9 | 17.9 | 52.0 | 20.5 | 87.3 | 2.1 | 29.8 | 54.9 | 62.5 | 37.9 |
| ♣Maskomaly (Ackermann et al., 2023) | **93.4** | **6.9** | - | - | - | - | 69.5 | 14.4 | 16.3 | 73.1 | 59.7 | 31.5 |
| ATTA (Gao et al., 2024) | 67.0 | 31.5 | 76.4 | 2.8 | 65.5 | 4.4 | **93.6** | **1.1** | 59.0 | 33.4 | 72.3 | 14.6 |
| FlowEneDet (Gudovskiy et al., 2023) | 36.7 | 77.8 | 73.7 | 0.9 | 56.1 | **3.8** | 66.6 | 8.9 | - | - | 58.3 | 22.9 |
| ♣♠Mask2Anomaly (Rai et al., 2023b) | 90.5 | 9.8 | 72.6 | 8.0 | 66.8 | 4.6 | 91.2 | 6.0 | 75.4 | 9.8 | 79.3 | 7.7 |
| ♣♠RBA (Nayal et al., 2023) | 92.3 | 7.2 | **92.4** | **0.3** | 50.4 | 10.8 | 69.5 | 6.0 | 83.5 | 18.7 | 77.6 | 8.6 |
| ▽♣♠DSA-Large (Ours) | 86.1 | 7.7 | 74.1 | 0.5 | **71.0** | 9.4 | 85.4 | 10.6 | 89.1 | 9.8 | **81.3** | **7.6** |

Table 3: **Quantitative Results:** We observe that on average our DSA-Large model obtains the best results among the baseline anomaly segmentation methods. The best and second best results are **bold** and underlined, respectively. ♣ shows the methods that are retrained. ♠ represents the methods using outlier exposure. ▽ method requires void labels while training.

## 5.1 MAIN RESULTS

We organize our results into two parts: (a) a comparative analysis of anomaly segmentation results between existing vision adapters and our dual-stream adapter, and (b) an evaluation that presents the efficacy of the dual-stream adapter by comparing it with all existing anomaly segmentation techniques.

**Adapter Comparison**

*Baseline setting:* Each adapter model consists of ViT-Tiny as backbone and mask-decoder. During the outlier supervision stage, we train all the adapter models with mask-contrastive loss (Rai et al., 2023b). We keep all the other implementation details the same as DSA-Tiny. More details about outlier supervision can be found in appendix A.1.

| Method | *without* Outlier supervision
mIoU | *with* Outlier supervision
mIoU |
|---|---|---|
| Mask2Former (Cheng et al., 2022) | 83.37 | - |
| ViT-Adapter (Chen et al., 2023) | 83.30 | - |
| RbA (Nayal et al., 2023) | 82.85 | 82.25 |
| EAM (Grcić et al., 2023) | 83.27 | 82.16 |
| DSA-Large (Ours) | **83.71** | **82.58** |

Table 4: **In-distribution performance**: In-distribution results on the Cityscapes validation set, for the vanilla Mask2Former and the recent anomaly segmentation SOTA based on it. DSA-Large retains the best in-distribution performance among these models. Best results are in **bold**.

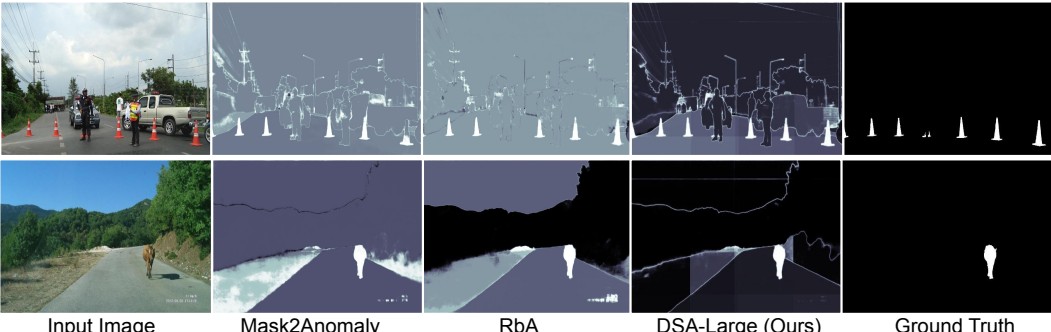

Input Image     Mask2Anomaly     RbA     DSA-Large (Ours)     Ground Truth

Figure 5: **Qualitative Results**: DSA-Large provide better and crisper anomaly masks w.r.t. other baselined methods. Anomalies are represented in white. The samples are shown for SMIYC RA-21 and SMIYC RO-21. Please refer to the appendix Fig. 8 for more results.

*Results discussion:* Table 1 and Table 2 display the performance of existing vision adapters and DSA-Tiny, with and without the outlier exposure setting, respectively. On average, DSA-Tiny shows significant performance improvement over existing vision transformer adapters by providing higher AuPRC values and lower FPR$_{95}$ scores, highlighting the effectiveness of the anomaly prior and the dual stream refinement. In the qualitative comparison shown in Fig. 4, we note that the vision transformer adapters struggle to segment anomalies, whereas DSA-Tiny demonstrates strong anomaly segmentation performance with minimal false positive rate. Additionally, it is worth noting that fine-tuning the entire architecture resulted in lower performance compared to our approach. This is in line with the findings of Kumar et al. (2022), who noted that fine-tuning leads to distorting the pre-trained features, thus, reducing anomaly detection performance.

**Evaluation with Anomaly Segmentation Methods**

*Baseline setting:* We establish consistency and fairness among all the anomaly segmentation methods by training them on Cityscapes (Cordts et al., 2016) as inlier dataset. For all the mask-transformer based anomaly segmentation methods (Rai et al., 2023b; Nayal et al., 2023; Ackermann et al., 2023; Grcić et al., 2023), we train them on Swin-L (Liu et al., 2021) without any additional inlier data as used in (Grcić et al., 2023).

*Results discussion:* **Anomaly Segmentation Performance**: Table 3 displays the performance of per-pixel and masked-based anomaly segmentation methods. We observe that in average DSA-Large obtains the best results among the recent SOTA mask-based architectures such as Mask2Anomaly (Rai et al., 2023b) and RbA (Nayal et al., 2023), although it appears that the results of the various methods vary depending on the datasets. Visually, from Fig. 5, we notice that mask-based methods RbA (Nayal et al., 2023) and Mask2Anomaly (Rai et al., 2023b) struggle to segment the anomalies.

**Semantic Segmentation Performance**: We also look at the performance of anomaly segmentation models on the in-distribution data. This is crucial, since some design choices made to boost the out-of-distribution detection may reduce the performance on the original semantic segmentation task, which may be a tradeoff not acceptable in practice (it is not practical to have two specialized models, one for ID and one for OOD). Table 4 reports the results on the Cityscapes validation dataset of the vanilla Mask2Former, as well as, anomaly segmentation methods based upon it. Notably, DSA-Large is able to achieve the best in-distribution accuracy, both when trained with and without

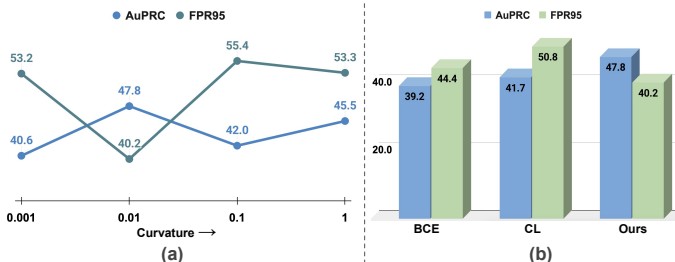

Figure 6: **(a)** Demonstrate the anomaly segmentation performance on different curvature values $c$. **(b)** Anomaly segmentation performances obtained by training DSA-Tiny on different loss functions. BCE, CL, and Ours refers to Binary Cross Entropy, Contrastive Loss, and $L_{ubhl}$, respectively.

| Anomaly Prior Module | Dual-Stream Feature Refinement | $L_{ubhl}$ | AuPRC↑ | FPR$_{95}$↓ |
|---|---|---|---|---|
| ✗ | ✓ | ✓ | 12.0 | 13.2 |
| ✓ | ✗ | ✓ | 18.5 | 7.7 |
| ✓ | ✓ | ✗ | 50.2 | 5.3 |
| ✓ | ✓ | ✓ | **59.6** | **3.0** |

| Number of Dual-Stream Adapter | AuPRC↑ | FPR$_{95}$↓ |
|---|---|---|
| 1 | 51.2 | 4.8 |
| 2 | 55.7 | 3.9 |
| 4 | **59.6** | 3.0 |
| 6 | 46.3 | **2.9** |

(a)  (b)

Table 5: (a) **Component wise ablation:** We test the importance of each component by removing one at a time. We observe that presence of all components gives the best performance. (b) **Number of dual-stream adapter:** We observe that having number of dual-stream adapter as 4 and 6 gives the best AuPRC and FPR$_{95}$, respectively. All the results in **bold** are best.

outlier supervision, confirming that our dual-stream adapter design strikes a good tradeoff between the two tasks of semantic segmentation and anomaly segmentation.

## 5.2 ABLATION STUDY

We conduct all ablation experiments on the DSA-Tiny architecture.

**Component-wise ablation:** Table 5(a) presents the component-wise ablation of the novel parts present in the dual-stream adapteron SMIYC RO-21. We find that removing the anomaly prior and dual-stream feature refinement modules shows a significant performance drop in anomaly segmentation. This is due to the reason that the network is not able to separate well ID and OOD features in the absence of the aforementioned components. Finally, using $L_{ubhl}$ further reduces the false positive rate and improves the AuPRC score.

**Number of dual-stream feature refinement:** In Tab. 5(b), we present the effect of the number of dual-stream feature refinement on anomaly segmentation performance on SMIYC RO-21. As observed from the table, setting the number of dual-stream feature refinement modules as 4 and 6 gives the best AuPRC and FPR95 scores, respectively. We opt for 4 modules to have fewer trainable parameters.

**Curvature size:** In Fig. 6(a), we discuss the anomaly segmentation performance with varying curvature size of the Poincaré ball. Results shown are the average performance taken on all the datasets. We empirically find that setting the curvature to 0.01 yields overall better performance compared to other values.

**Loss comparison:** Figure 6(b) shows the anomaly segmentation performance comparison when using our proposed loss $L_{ubhl}$, and when replacing it with a binary cross-entropy loss or contrastive loss. The results are the average performance given by all datasets. We find that $L_{ubhl}$ yields the best performance among all the losses, confirming the advantage of using the hyperbolic representation to separate the features.

## 6 CONCLUSION

In this work, we introduce a novel adapter architecture for anomaly segmentation. Our architecture incorporates two key architectural components: anomaly prior modules and dual-stream feature refinement tailored to improve anomaly segmentation. The anomaly prior module learns to extract the initial ID and OOD features that are refined and improved by passing through a set of dual-stream feature refinement blocks. We introduce an uncertainty-based hyperbolic loss to explicitly learn the ID and OOD features. We hope that our dual-stream adapter will pave new ways for developing efficient anomaly segmentation methods based on vision adapters.

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

# A   APPENDIX

## A.1   IMPLEMENTATION DETAILS

**DSA-Tiny:** The model architecture consists of a ViT-Tiny the backbone and a Mask2Former decoder, which includes 9 mask attention layers. The ViT backbone is pre-trained on ImageNet-1K. The number of dual-stream adapter blocks are kept 4. The FFN consists of two MLP layers having a GELU activation layer between them. We used deformable attention architecture as a cross-attention in our architecture. We used the AdamW optimizer with an initial learning rate of $12 \times 10^{-5}$, a batch size of 32, and a weight decay of 0.01. Layer-wise learning rate decay of 0.95 is used during training. The DSA-Tiny model was trained for 46000 iterations on Cityscapes, utilizing a batch size of 32 and an image crop size of $512 \times 512$. The model contains a total of 28.5M parameters, with 5.5M from the frozen ViT backbone. Of the remaining 23M trainable parameters dual-stream adapter has 2.4M parameters.

**DSA-Large:** The architecture of DSA-Large consists of a ViT-Large backbone with BEiT architecture and a Mask2Former decoder, which includes two mask attention layers. The ViT backbone is pre-trained on ImageNet-22K dataset. The number of dual-stream adapter blocks is set to 4. We employed the AdamW optimizer with an initial learning rate of $2 \times 10^{-5}$ and a weight decay of 0.05. During training, we applied layer-wise learning rate decay of 0.9. The DSA-Large model was trained for $60,000$ iterations on Cityscapes, with a batch size of 8 and an image crop size of $768 \times 768$. The model has a total of 435.9M parameters, of which 303.3M are from the frozen ViT backbone. Out of the remaining 133M trainable parameters, dual-stream adapter accounts for 27.7M parameters.

**Outlier supervision** is a widely used technique to improve the performance of anomaly segmentation methods (Rai et al., 2023b; Nayal et al., 2023; Grcić et al., 2022) by training models with synthetic anomaly images generated on the fly. In our approach, we create these synthetic images using AnomalyMix (Tian et al., 2022), which cut objects from the MS-COCO (Lin et al., 2014) dataset based on ground truth annotations and randomly pasting them onto Cityscapes images. This produces a binary mask, where the MS-COCO object area is labeled as 1 showing an anomalous region, and the Cityscapes background area is labeled as 0 for the non-anomalous region. For training, we employ a mask contrastive loss (Rai et al., 2023b). During this phase, only the mask decoder is trained, and the loss term $L_{ubhl}$ is not applied since DSA is frozen. We train DSA-Tiny and DSA-Large for 2000 iterations, using batch sizes of 8 and 2, respectively. The probability of selecting an outlier in each training batch is set to 0.2. All other hyperparameters are kept consistent with previous settings. The margin for the mask contrastive loss is set to 1.0.

## A.2   DATASETS AND EVALUATION METRICS

We evaluate all anomaly segmentation methods using the Area under the Precision-Recall Curve (AuPRC), which is well-suited for imbalanced datasets. This makes it particularly suited for anomaly segmentation, where datasets are highly skewed. Additionally, we use the False Positive Rate at a True Positive Rate of 95% ($\text{FPR}_{95}$) as a key metric, especially important for safety-critical applications. Next, we use the Road Anomaly Lis et al. (2019), Fishyscapes Blum et al. (2021), and Segment Me If You Can (SMIYC) benchmarks Chan et al. (2021a) as evaluation dataset discribed below.

- Road Anomaly consists of 60 images containing anomalous objects on or near the road that are taken from the web.
- Fishyscapes (FS) includes two datasets: Fishyscapes Static (FS Static) and Fishyscapes Lost & Found (FS L&F). FS Static is created by blending Pascal VOC objects into Cityscapes images and contains 30 validation images. FS L&F is based on a subset of the Lost and Found dataset with 100 validation images.
- SMIYC consists of two datasets: RoadAnomaly21 (SMIYC-RA21) and RoadObstacle21 (SMIYC-RO21). SMIYC-RA21 has 10 validation images and 100 test images with diverse anomalies. SMIYC-RO21 is focused on segmenting road anomalies and contains 30 validation images and 327 test images.

Figure 7: **Qualitative Ablation:** We demonstrate the importance of DSA components by individually removing them one at a time. DSFI and APM stand for dual-stream feature refinement and anomaly prior, respectively. Anomalies are represented in white.

## A.3 ADDITIONAL DETAILS FOR ABLATION STUDY AND RESULTS

| | SMIYC RA-21 | | SMIYC RO-21 | | FS L&F | | FS Static | | Road Anomaly | |
|---|---|---|---|---|---|---|---|---|---|---|
| | AuPRC ↑ | FPR$_{95}$ ↓ | AuPRC ↑ | FPR$_{95}$ ↓ | AuPRC ↑ | FPR$_{95}$ ↓ | AuPRC ↑ | FPR$_{95}$ ↓ | AuPRC ↑ | FPR$_{95}$ ↓ |
| *with* injector | 57.1 | 90.7 | **60.3** | **7.5** | 42.5 | 27.0 | **31.5** | **33.1** | **26.0** | 88.3 |
| *without* injector | **62.9** | **76.4** | 33.9 | 11.2 | **46.7** | **21.1** | 27.1 | 90.2 | 25.3 | **81.7** |

Table 6: Anomaly segmentation performance with and without injector. Best performance in **bold**

**Role of injector:** The ViT Adapter Chen et al. (2023) includes an injector designed to integrate spatial priors into the ViT blocks. However, as demonstrated empirically in Tab. 6, the removal of the injector had no significant impact on anomaly segmentation performance. Based on this observation, we excluded the injector from our architecture, reducing the model's complexity and saving parameters. For the experiment, we choose ViT tiny backbone of the adapter.

| | SMIYC RA-21 | | SMIYC RO-21 | | FS L&F | | FS Static | | Road Anomaly | |
|---|---|---|---|---|---|---|---|---|---|---|
| Curvature (c) | AuPRC ↑ | FPR$_{95}$ ↓ | AuPRC ↑ | FPR$_{95}$ ↓ | AuPRC ↑ | FPR$_{95}$ ↓ | AuPRC ↑ | FPR$_{95}$ ↓ | AuPRC ↑ | FPR$_{95}$ ↓ |
| 1.0 | 54.8 | **77.4** | **64.4** | **2.8** | **50.6** | 20.9 | 25.0 | 78.1 | 32.4 | 87.1 |
| 0.1 | 55.5 | 81.7 | 44.0 | 2.9 | 45.7 | 24.2 | 32.8 | 89.6 | 31.5 | 78.4 |
| 0.01 | **58.3** | 80.4 | 59.6 | 3.0 | 44.0 | 20.0 | **40.1** | **19.9** | **36.7** | **77.3** |
| 0.001 | 56.8 | 83.1 | 48.4 | 4.9 | 44.4 | **19.3** | 25.4 | 67.3 | 27.8 | 90.8 |

Table 7: Anomaly segmentation performance on varying curvature of poincaré ball. Best performance in **bold**

**Curvature of poincaré** ($c$)**:** Table 7 presents the anomaly segmentation performance across different curvature values of the Poincaré ball. Notably, no single curvature value consistently gives the best performance across all datasets. However, on average, a curvature of 0.01 yields the most favorable results. All experiments were conducted on DSA-Tiny, with $\lambda_{ubhl}$ set to 0.1.

| | SMIYC RA-21 | | SMIYC RO-21 | | FS L&F | | FS Static | | Road Anomaly | |
|---|---|---|---|---|---|---|---|---|---|---|
| Loss | AuPRC ↑ | FPR$_{95}$ ↓ | AuPRC ↑ | FPR$_{95}$ ↓ | AuPRC ↑ | FPR$_{95}$ ↓ | AuPRC ↑ | FPR$_{95}$ ↓ | AuPRC ↑ | FPR$_{95}$ ↓ |
| Binary Cross Entropy | 50.8 | 85.6 | 50.2 | 5.3 | 39.3 | 21.0 | 27.3 | 21.4 | 27.9 | 88.5 |
| Contrastive Loss | 47.6 | 84.6 | **60.4** | 3.6 | 43.5 | 43.3 | 32.5 | 32.7 | 24.1 | 89.6 |
| $L_{ubhl}$ | **58.3** | **80.4** | 59.6 | **3.0** | **44.0** | **20.0** | **40.1** | **19.9** | **36.7** | **77.3** |

Table 8: Loss Comparison among our proposed loss and euclidean losses. Best performance in **bold**

**Loss comparison:** In Tab. 8, we show that $L_{ubhl}$ achieves the best performance across all datasets. All experiments were conducted on DSA-Tiny. While using the contrastive loss a margin of 1.0 is kept. The loss weight in the total loss (10) was set to 0.1 for all comparisons.

| $\lambda_{ubhl}$ | SMIYC RA-21 | | SMIYC RO-21 | | FS L&F | | FS Static | | Road Anomaly | |
|---|---|---|---|---|---|---|---|---|---|---|
| | AuPRC $\uparrow$ | FPR$_{95}$ $\downarrow$ | AuPRC $\uparrow$ | FPR$_{95}$ $\downarrow$ | AuPRC $\uparrow$ | FPR$_{95}$ $\downarrow$ | AuPRC $\uparrow$ | FPR$_{95}$ $\downarrow$ | AuPRC $\uparrow$ | FPR$_{95}$ $\downarrow$ |
| 2.0 | **58.2** | 81.5 | 63.5 | 5.0 | 43.4 | 23.9 | 25.2 | 46.0 | **33.6** | 79.2 |
| 1.0 | 50.8 | 88.8 | 43.7 | 7.6 | 47.9 | 25.1 | **28.8** | **23.4** | 29.2 | 85.1 |
| 0.1 | 54.9 | **77.4** | **64.5** | 2.9 | **50.6** | **20.9** | 25.1 | 78.2 | 32.4 | 87.1 |
| 0.01 | 56.1 | 83.6 | 42.3 | **2.2** | 46.1 | 44.1 | 22.7 | 26.0 | 27.7 | **78.2** |

Table 9: Variation in anomaly segmentation performance with different $\lambda_{ubhl}$. Best performance in **bold**.

**Variation in $\lambda_{ubhl}$:** Table 9 shows the anomaly segmentation performance for different values of $\lambda_{ubhl}$ as defined in (10). We observe no single value of $\lambda_{ubhl}$ consistently outperforms across all datasets, but on average, a value of 0.1 yields the best results. All experiments were conducted on DSA-Tiny.

**Qualitative ablation study:** In Tab. 5, we present the quantitative results of the component-wise ablation. The quantitative behavior can also be corroborated qualitatively from Fig. 7 where we observe significant increase in false positives rates when anomaly prior and dual-stream feature refinement are removed.

**Comparision with UNO**: UNO Delić et al. (2024) uses Mapillary Vistas as additional inlier data to boost anomaly segmentation performance and ADE-20K as outlier exposure dataset. To ensure fairness and consistency, we use the same training protocol as in our draft. We trained UNO using Cityscapes as the inlier dataset and incorporated MS-COCO image objects during the outlier-exposure phase. The table below presents the performance comparison across all validation datasets. We can observe that DSA-Large consistently outperform UNO across all datasets.

| Method | SMIYC RA-21 | | SMIYC RO-21 | | FS L&F | | FS Static | | Road Anomaly | |
|---|---|---|---|---|---|---|---|---|---|---|
| | AuPRC | FPR$_{95}$ | AuPRC | FPR$_{95}$ | AuPRC | FPR$_{95}$ | AuPRC | FPR$_{95}$ | AuPRC | FPR$_{95}$ |
| UNO | 62.7 | 33.1 | 12.5 | 25.6 | 35.0 | 42.8 | 82.3 | 12.2 | 54.1 | 17.0 |
| DSA-Large | 73.3 | 45.3 | 92.6 | 0.72 | 71.0 | 9.4 | 85.4 | 10.6 | 89.1 | 9.8 |

Table 10: Performance metrics comparison across different methods and datasets.

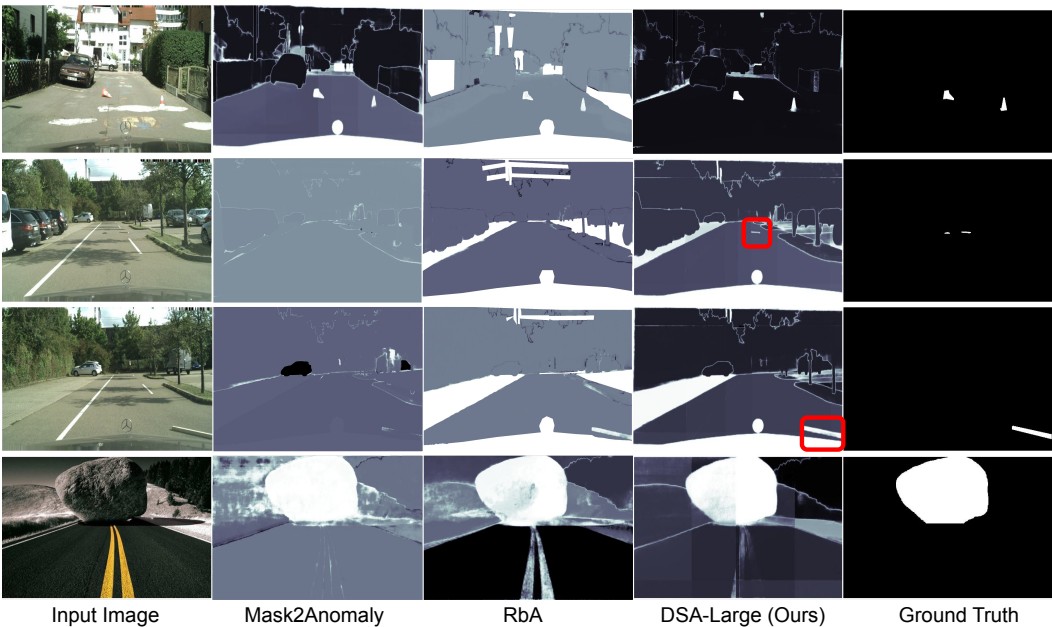

| Input Image | Mask2Anomaly | RbA | DSA-Large (Ours) | Ground Truth |

Figure 8: **Additional qualitative results:** *Row 1 and Row 4*: Displays that DSA-large is able to segment anomalies with minimal false positives. *Row 2 and Row 3*: shows that anomalies that are left unsegmented by Mask2Anomaly and RbA are well segmented by DSA-large as shown in red bounding box.

