# OpenReview forum: "Dual-Stream Adapters for Anomaly Segmentation"
_ICLR.cc/2025/Conference — Submitted to ICLR 2025_

### Official Review · Reviewer_Wr5a · 2024-10-23

**Soundness:** 2
**Presentation:** 3
**Contribution:** 1
**Rating:** 3
**Confidence:** 4

**Summary:**

This paper introduces a novel adapter architecture for anomaly segmentation. The architecture incorporates two key architectural components: anomaly prior modules and dual-stream feature refinement tailored to improve anomaly segmentation. The anomaly prior
module learns to extract the initial ID and OOD features that are refined and improved by passing through a set of dual-stream feature refinement blocks. It introduces an uncertainty-based hyperbolic loss to explicitly learn the ID and OOD features. The results on several datasets in terms of some metrics show the effectiveness.

**Strengths:**

+ a novel adapter architecture for anomaly segmentation. It designs anomaly prior modules and dual-stream feature refinement tailored to improve anomaly segmentation.
+ The results on several datasets in terms of some metrics are the SOTA.

**Weaknesses:**

- The definaton of this task is unclear. How to define the Anomaly regions. For example, I put the first input of Fig. 5 into the ChatGPT and ask it to indicate the abnormal regions. It responses: "
From the image, it appears to show a traffic checkpoint or roadblock scene with a few police officers, some cones marking the road, and vehicles either being stopped or passing through the checkpoint. To analyze any potential abnormal regions, I would need specific criteria for identifying abnormalities, such as:
Traffic flow: Are there any unusual movements or traffic behavior?
Objects: Are there any misplaced cones, vehicles, or people?
Security concerns: Is there anything out of place, like suspicious behavior or unmarked vehicles?
If you're looking for more detailed insights, could you clarify what type of abnormality you're focused on—whether it's safety, traffic irregularities, or something else?"
This answer is more reasonable, since the anomaly regions can be different in different settings.

-The above example gives another question. The current multimodal large language model already achieves better responses.

- The preposed method lacks novelty, and it is mainly combined by Vit adaptors with cross-attention.

**Questions:**

Based on the above the weaknesses, I have the following questions:

- What are the key criteria or benchmarks used to evaluate anomaly detection in your scenario?
Defining the task clearly with precise criteria is critical. How do you plan to measure the model's success in detecting abnormal regions?

- What are the distinguishing features of your method compared to existing multimodal models?

- How does the model handle the contextual differences in anomaly definitions?
Since anomalies are highly context-dependent, is your model trained to recognize such context? How does it deal with shifting or ambiguous definitions of normalcy versus anomaly?

- Are there plans for improving the generalization and adaptability of the model?
For instance, using domain adaptation techniques or fine-tuning the model on specific datasets related to the task could improve its performance in different scenarios.

---

> ### Author Response · Authors · 2024-11-26
>
> **Q1, Q3, and W1 - task definition, metrics, and benchmarks:** We agree with the reviewer that the concept of anomaly is very broad and open to different interpretations depending on the context and field of application. In this work, we consider specifically the task of anomaly segmentation in driving scenes, a task with a precise definition adopted by a plethora of works [A,B,C,D,E,F,G,H,I,J,K,L,M,N]. In autonomous driving applications, image segmentation is an important functionality that aims to precisely classify all the pixels according to a set of in-distribution categories that a model is exposed to during training. In this context, anomaly segmentation aims to detect and locate, with pixel-level granularity, objects belonging to previously unseen categories that are present in road scenes. This task definition is supported by several well-established benchmarks: Fishyscapes [K], Segment Me If You Can [J], Road Anomaly [M],  StreetHazards [L], and BDD-Anomaly [L]. All these benchmarks assume a closed set of in-distribution categories, typically following the Cityscapes taxonomy (e.g., car, pedestrian, bicycle, etc.), and the anomalies are out-of-distribution categories that do not belong to this taxonomy (e.g., wildlife).
> The metrics used across these benchmarks to measure the model performance, as mentioned in Section 5 and in Appendix A.2, are $AuPRC$ and $FPR_{95}$.   Using these two metrics is a common practice in this line of research [J,K,M,L], as they provide a good insight into different aspects: the AuPRC puts emphasis on detecting the minority class and it is generally considered to evaluate the separability of the pixel-wise anomaly scores, whereas the $FPR_{95}$ metric indicates how many false positive predictions must be made to reach the desired true positive rate. A well-performing anomaly segmentation will have high $AuPRC$ with lower $FPR_{95}$.
> We recognize that the terminology anomaly segmentation may be misleading, but we simply adhered to the terminology already in use in this field. To make the task definition and context more clear, we rephrased the lines between 028-030 in the introduction.
>
> **W2 and Q2 - multimodal large language models:** It is true that multimodal large language models have shown impressive performance on general computer vision tasks, particularly tasks that require interactivity (e.g., task prompted segmentation) or reasoning. However,  the task of anomaly segmentation for road scenes, as defined above, is a domain-specific task that requires a separation of the in-distribution categories (as provided by a training dataset) and out-of-distribution elements. It is also a task that does not have an interactive user component, which may warrant textual prompting. This is a task that can be addressed effectively by models that are specifically trained for the domain, and ours is particularly capable not only of segmenting anomalies but also retaining excellent in-distribution performance. Moreover, relying on an additional model, such as a multimodal LLM, besides the segmentation model deployed to segment the in-distribution classes is not a practical solution. Indeed, to the best of our knowledge, **there are no works that address anomaly segmentation in driving scenes using a multimodal large language model**, as suggested by the Reviewer.
> We are only aware of one arxiv preprint (currently listed as work in progress) titled _Towards Generic Anomaly Detection and Understanding: Large-scale Visual-linguistic Model (GPT-4V) Takes the Lead_, which provides a first qualitative exploration on the use of GPT-4V for anomaly detection. However, that method **does not provide pixel-level predictions of anomalies**.
> The level of reasoning that can be achieved by multimodal LLM can be very useful when moving towards a general formulation of anomaly detection in driving scenes, as in the examples given by the Reviewer. Yet, that is not the aim of this paper and we are not aware of any benchmarks or datasets that currently support that kind of more general research in the driving setting.

---

> > ### Author Response · Authors · 2024-11-26
> >
> > **W3 - novelty:** As the Reviewer recognizes (in **S1**), our paper contributes “_a novel adapter architecture for anomaly segmentation_”. Indeed, although several adapter methods have been presented for different computer vision tasks, **none of them are specifically intended for anomaly segmentation tasks**. Our dual-stream adapter is specifically designed for that purpose, thus differing from other more general  ViT adapters. Please, also note that the novelty of our design lies not only in the architectural components of the adapter but also in the training procedure, using an uncertainty-based hyperbolic loss. The various components that we introduce in our design are shown to be beneficial for their intended purpose (anomaly segmentation):
> > We propose an *anomaly prior module* that learns in-distribution and out-of-distribution features through distinct feature-level encodings. Table 5 (a) shows utilizing the anomaly prior module significantly improves the anomaly segmentation performance.
> > We introduce a *dual-stream feature refinement module* that improves the in-distribution and out-of-distribution features by combining frozen ViT features. In  Table 5 (a), we can observe that the absence of a dual-stream feature refinement module drastically lowers the anomaly segmentation performance.
> > We presented *uncertainty-based hyperbolic loss* that makes the network separate the in-distribution with out-of-distribution features in hyperbolic space. Figure 6 (b) and Table 5 (a) show the effectiveness of this loss.
> >
> > **Q4 - generalization and adaptability:** Indeed, the problem of the generalization and adaptability of anomaly segmentation methods for driving scenes is still an open research direction. We observed a certain level of variability of ours and other methods across different datasets, due to the domain gap between training and test domain. As also shown in Table 3, there is not a single method performing the best on all datasets, possibly due to this problem. Addressing such a domain shift is an interesting direction, one that we will try to pursue in the future. However, the currently available datasets are relatively small and are not designed to analyze this problem, so it will be necessary to establish a new benchmark suitable for this purpose, with well-defined training and testing protocols.
> >
> > [A] Zhitong Gao, Shipeng Yan, and Xuming He. Atta: Anomaly-aware test-time adaptation for out-of-distribution detection in segmentation, NeurIPS 2024.
> >
> > [B] Shyam Nandan Rai, Fabio Cermelli, Barbara Caputo, and Carlo Masone. Mask2anomaly: Mask transformer for universal open-set segmentation. T-PAMI 2024
> >
> > [C] Nazir Nayal, Misra Yavuz, Joao F Henriques, and Fatma Güney. Rba: Segmenting unknown region rejected by all. ICCV 2023
> >
> > [D] Matej Grcic ́, Josip Sˇaric ́, and Sinisˇa Sˇegvic ́. On advantages of mask-level recognition for outlier- aware segmentation. CVPRw 2023.
> >
> > [E] Denis Gudovskiy, Tomoyuki Okuno, and Yohei Nakata. Concurrent misclassification and out-of- distribution detection for semantic segmentation via energy-based normalizing flow, UAI 2023
> >
> > [F] Jan Ackermann, Christos Sakaridis, and Fisher Yu. Maskomaly: Zero-shot mask anomaly segmentation. In 34th British Machine Vision Conference 2022, BMVC 2022.
> >
> > [G] Matej Grcic ́, Petra Bevandic ́, and Sinisˇa Sˇegvic ́. DenseHybrid: Hybrid anomaly detection for dense open-set recognition, ECCV 2022.
> >
> > [H] Robin Chan, Matthias Rottmann, and Hanno Gottschalk. Entropy maximization and meta classification for out-of-distribution detection in semantic segmentation, ICCV 2021.
> >
> > [I] Giancarlo Di Biase, Hermann Blum, Roland Siegwart, and Cesar Cadena. Pixel-wise anomaly detection in complex driving scenes, CVPR 2021.
> >
> > [J] Robin Chan, Krzysztof Lis, Svenja Uhlemeyer, Hermann Blum, Sina Honari, Roland Siegwart, Pascal Fua, Mathieu Salzmann, and Matthias Rottmann. SegmentMeIfYouCan: A benchmark for anomaly segmentation. NeurIPS 2021
> >
> > [K] Hermann Blum, Paul-Edouard Sarlin, Juan Nieto, Roland Siegwart, and Cesar Cadena. The Fishyscapes benchmark: Measuring blind spots in semantic segmentation. IJCV, 2021
> >
> > [L] Dan Hendrycks, Steven Basart, Mantas Mazeika, Andy Zou, Joe Kwon, Mohammadreza Mostajabi, Jacob Steinhardt and Dawn Song. Scaling Out-of-Distribution Detection for Real-World Settings. ICML 2022
> >
> > [M] Krzysztof Lis, Krishna Nakka, Pascal Fua and Mathieu Salzmann. Detecting the Unexpected via Image Resynthesis. ICCV 2019
> >
> > [N] Matteo Sodano, Federico Magistri, Lucas Nunes, Jens Behley and Cyrill Stachniss. Open-World Semantic Segmentation Including Class Similarity. CVPR 2024

---

### Official Review · Reviewer_R2z4 · 2024-10-31

**Soundness:** 3
**Presentation:** 3
**Contribution:** 3
**Rating:** 5
**Confidence:** 4

**Summary:**

The paper aims to reduce the high training cost of the transformer-based anomaly segmentation model. The authors introduce the Dual-Stream Adapter (DSA) to tackle this issue, specifically designed for anomaly segmentation tasks. The DSA comprises two main components: an anomaly prior module and a dual-stream feature refinement module. The anomaly prior module is responsible for learning both in-distribution and out-of-distribution features through separate feature-level encodings. Meanwhile, the dual-stream feature refinement module enhances and integrates these features with the robust representations derived from a frozen Vision Transformer backbone. Additionally, the architecture employs a hyperbolic loss function to guide the learning process effectively. The authors carried out extensive experiments across five datasets, demonstrating their proposed model's effectiveness.

**Strengths:**

+ The paper is clearly structured, featuring well-organized paragraphs and accessible figures that enhance comprehension.
+ The design of the proposed Dual-Stream Adapter is sound and well-motivated.

**Weaknesses:**

- The experiments conducted were compared to several competitors; however, the values reported in Table 3 do not align with those in the original publications. This inconsistency raises questions about the validity of the experimental outcomes. It is highly advisable for the authors to verify the accuracy of the reported experiments to ensure reliable results.

- It appears that there is an inconsistency between Figure 3 and equations (6) and (7). It would be beneficial to verify their accuracy.

- It would be interesting to analyze the amount of training time that can be saved due to the reduction in trainable parameters.

- There are some related papers [A-B] that have not yet been cited in the current version.

    [A]  Yuanpeng Tu, Yuxi Li, Boshen Zhang, Liang Liu, Jiangning Zhang, Yabiao Wang, Cairong Zhao: Self-Supervised Likelihood Estimation with Energy Guidance for Anomaly Segmentation in Urban Scenes. AAAI 2024: 21637-21645

    [B] Tomás Vojír, Jirí Matas: Image-Consistent Detection of Road Anomalies as Unpredictable Patches. WACV 2023: 5480-5489

**Questions:**

This paper is well-structured and has a clear motivation. However, a significant concern arises from the experiments reported, as the values presented appear inconsistent with those in the original publications. At this stage, I prefer to take a cautious approach to the rating and await the author's response before reaching a final conclusion.

---

> ### Author Response · Authors · 2024-11-26
>
> **W1 - experimental results:** The Reviewer is correct in observing that the competitors' results in Table 3 are not the same as those in their original publications, and this is not by mistake. In fact, one of the biggest problems we have encountered in this line of research and in benchmarks such as SMIYC is that even though the testing protocol is well-defined, the training protocol used by different methods can vary a lot, resulting in unfair and not informative comparisons. For instance, if one checks carefully the methods in the SMIYC leaderboard would find that there are a lot of discrepancies among them:
> Additional inlier datasets: there are methods (e.g. UNO) that instead of training solely on Cityscapes, leverage additional inlier data from datasets such as Mapillary Vistas, which can significantly enhance anomaly segmentation performance.
> Variation in backbones: Different backbones are used in these methods—for instance, RbA uses Swin-B while EAM utilizes Swin-L. It was shown that using larger backbones often leads to improved performance [B, C], thus such unregulated comparisons are unfair.
> Outlier Exposure: It has been observed the choice of outlier exposure datasets like ADE20K or MS-COCO can significantly vary the model’s anomaly segmentation performance.
> These are just a few of the things we observed and we believe that this kind of _apples to oranges_ comparison is unfair and may hide the real contribution of different methods. For this reason, for all most recent competitors using mask-based architectures (which are the current SOTA), we did not simply report the results from their papers, but we re-evaluated them following the same training protocol for all methods, using Cityscapes as inlier dataset, Swin-L as the backbone, and MS-COCO as the choice of outlier exposure datasets. We did not stress this as a contribution because it is not the goal of the paper to establish a fair benchmark, but we firmly believe that this kind of rigorous evaluation is necessary and valuable for this field. To make things more clear in the paper, we added a symbol in Table 3 and further details between lines 468-473 to indicate which methods have been re-trained.
>
> **W2 - Fig3 & Eq 6,7:** We thank the reviewer for pointing out the discrepancy. We have changed the equation and figure accordingly.
>
> **W3 - training time:** Our dual-stream adapters take ~65 Hours for the full training of the network. On the other hand, without the adapters, the full finetuning of the network requires ~77 hours of training time.
>
> **W4 - missing citations:** Thank you for suggesting the citations. _Image-Consistent Detection of Road Anomalies as Unpredictable Patches_ proposes DaCUP, a method that uses a novel embedding bottleneck and image-conditioned distance features to improve anomaly segmentation. _Self-Supervised Likelihood Estimation with Energy Guidance for Anomaly Segmentation in Urban Scenes_ introduces two estimators to model anomaly likelihood where one is a task-agnostic binary estimator and the other gives the likelihood as residual of task-oriented joint energy.  Differently from ours, these works are based on per-pixel architectures that are not well suited for anomaly segmentation and they require the entire network to be fine-tuned. We have added them to the related work of the draft.

---

> ### Comment · Reviewer_R2z4 · 2024-11-27
> **Respond to the Rebuttal**
>
> Thanks to the author's effort in providing a rebuttal, as it has offered me additional information. I would like to finalize my scoring after checking the comments from other reviewers.

---

### Official Review · Reviewer_wmG8 · 2024-11-04

**Soundness:** 3
**Presentation:** 3
**Contribution:** 3
**Rating:** 6
**Confidence:** 4

**Summary:**

This paper introduces a ViT adapter designed specifically for anomaly detection in semantic segmentation. The approach begins by extracting multi-scale features using a ResNet convolutional stem. Two distinct sets of learnable encodings are added to these features to specialize them for in-distribution (ID) and out-of-distribution (OOD) learning. These initial features are then fused with ViT backbone activations through a series of cross-attention layers and feed-forward networks, maintaining separate streams for ID and OOD features throughout the entire downstream path. The model is trained using an uncertainty-based hyperbolic loss function, where features are projected into hyperbolic space: OOD features are attracted towards the origin of the Poincaré ball, while ID features are repelled away from it.

The authors evaluate their dual-stream adapter on road-driving anomaly detection benchmarks, comparing its performance to other ViT adapters and alternative outlier detection methods. Additionally, they conduct an ablation study to assess the impact of individual components of their proposed method.

**Strengths:**

1) Method is simple and more parameter efficient than existing outlier detection methods.
2) The method preserves ID performance well.

**Weaknesses:**

1) Mixed results in terms of OOD detection on different datasets, both in comparison to fully fine-tuned models, but also different adapter types.
2) I do not agree that using void/background class makes this training self-supervised, as there is explicit supervision in terms of labelling pixels into this additional class. Furthermore, the method could not be used on datasets that do not have ignore regions. It would maybe be better to follow the convention of using the term "auxiliary data". It  would be useful if this was reflected in the organization of Table 3.
3) Missing methods in ood-detection tables. e.g. [a] is the current leader on SMIYC-RA
4) Some things about the method are still not clear -> see questions.

[a] Delic et al., Outlier detection by ensembling uncertainty with negative objectness, BMVC 2024

**Questions:**

1) How are the encoder and adapter features integrated before being passed to the decoder as described in Equation 3?
2) Is the background class utilized in the computation of L_{mask} and L_{class}​?
3) Is the convolutional module within the anomaly prior module also fine-tuned? Was it pretrained, and if so, how? What specific ResNet variant is used?
4) Can you provide detailed specifications of the Cross-Attention and Feed-Forward Network (FFN) modules used in the adapters?
5) How is the void/background class incorporated into L_{ubhl}? At what stage is the ground truth label applied?

---

> ### Author Response · Authors · 2024-11-26
>
> **W1 - Mixed results:** We thank the Reviewer for the attentive observation. We observed that there is a certain variability among different datasets, not just for our method but also for the competitors. Indeed, the results in Table 3 show that there is no single method that is the best in all datasets. We conjecture that this variation in performance is to be attributed to two primary factors:  a) the size of the evaluation datasets and b) the domain shift between the training and evaluation datasets. The presence of such attributes among the anomaly evaluation datasets makes a method perform well on certain anomaly datasets while underperforming on others. For instance, methods like Mask2Anomaly perform exceptionally well on the SMIYC RA-21 dataset but fail to segment anomalies effectively on the SMIYC RO-21 dataset. Maskomaly is the best on SMIYC RA-21, but gives much lower results on FS-Static. A similar observation holds for the comparative analysis between various adapter types and anomaly segmentation methods presented in Table 1,2,3. We note that no method performs best in all datasets.
>
> For these reasons, we believe that it is important to look at the average performance across all datasets. In average, the dual-stream adapter demonstrates the best performance across all anomaly segmentation methods and adapter types, being the most robust to the different types of benchmarks and anomalies.
>
> **W2 - Self-supervision:** We appreciate the reviewer’s insightful observation. We used the term self-supervised to indicate that we train our model without requiring another dataset aside from the one used to train for the in-distribution categories, i.e., Cityscapes. However, this may indeed be inaccurate since we assume the presence of a void/background class to provide explicit supervision to the network. So, we will follow the advice and  remove the term"self-supervised" in our revised draft. In the revised draft we write that our method uses void/background labels as supervision. We think it is best to not refer to the void/background class as auxiliary data to avoid confusion with respect to the existing literature, because in the context of anomaly segmentation "auxiliary data" typically refers to additional datasets, such as MS-COCO or ADE-20K, whose objects are used during the outlier-exposure phase. In contrast, the supervision in our case comes from the same dataset, i.e., Cityscapes. To avoid any confusion between these two kinds of supervision, in Table 3 we have introduced two symbols: one to indicate the methods that require void labels for training (our method), and another symbol to denote the methods that utilize extra data to perform outlier-exposure.
>
> **W3 - Missing method:** Thank you for highlighting the overlooked method. We observe that UNO [a] uses Mapillary Vistas as additional inlier data to boost anomaly segmentation performance and ADE-20K as an outlier exposure dataset. To ensure fairness and consistency, we use the same training protocol as in our draft. Namely, we trained UNO using Cityscapes as the inlier dataset and incorporated MS-COCO image objects during the outlier-exposure phase. The table below presents the performance comparison across all validation datasets. We can observe that DSA-Large consistently outperforms UNO across all datasets. All the results are on the validation dataset due to the limited rebuttal time duration. We have included the results in the manuscript's appendix.
>
>
>
> |            |      SMIYC RA-21  ||  SMIYC RO-21  || FS L\&F  || FS Static  || Road Anomaly ||
> |--------------------|---------|---------|---------|---------|---------|---------|---------|---------|---------|----------|
> | Method                  | AuPRC | FPR$_{95}$ |  AuPRC| FPR$_{95}$ |  AuPRC | FPR$_{95}$ |  AuPRC | FPR$_{95}$ |  AuPRC | FPR$_{95}$ |
> | UNO                       | 62.7    | 33.1    | 12.5    | 25.6    | 35.0    | 42.8    | 82.3    | 12.2    | 54.1    | 17.0     |
> |  DSA-Large  | 73.3    | 45.3    | 92.6    | 0.72    | 71.0    | 9.4     | 85.4    | 10.6    | 89.1    | 9.8      |

---

> > ### Author Response · Authors · 2024-11-26
> >
> > **Q1- Equation 3 :** We integrate the encoder and adapter features with the help of a feature interaction module. In Dual-Stream adapters, dual stream feature refinement module is utilized for fusing the features. We have added these details between lines 762-764.
> >
> > **Q2 - Background Class:** No, we only utilize in-distribution classes while computing $L_{mask}$ and $L_{class}$.
> >
> > **Q3 - Anomaly Prior Module:** Yes, the convolutional module within the anomaly prior module is trained before the outlier exposure step. During the outlier exposure step, the entire anomaly prior module is kept frozen. The details can be found between lines 785-789.
> >
> > **Q4 - Cross-Attention and Feed-Forward Network specifications:** The FFN consists of two MLP layers having a GELU activation layer between them. We used deformable attention architecture as a cross-attention in our architecture.  We have clarified these details in the manuscript's appendix.
> >
> > **Q5 - $L_{ubhl}$ :** Thank you for pointing out the missing details. In $L_{ubhl}$, we use the background/void class as the anomalous region. We added details in Equation 9 for better clarity.

---

> > > ### Comment · Reviewer_wmG8 · 2024-12-02
> > > **Final score**
> > >
> > > I would like to thank the authors for the clarifications.
> > >
> > > I have therefore decided to increase my rating.

---

### Author Response · Authors · 2024-12-03

We thank all the reviewers for their suggestions and comments. In our responses, we tried to address all the concerns. Please, let us know if any questions should remain, we are available for further clarification.

---

### Meta-Review · Area_Chair_Mrhs · 2024-12-16

**Metareview:**

The paper proposes a ViT adapter for anomaly detection in semantic segmentation. The proposal uses two streams for in-distribution and out-of-distribution, respectively. The model is trained using an uncertainty-based hyperbolic loss.

Strengths:
- Simple method
- Uses fewer parameters than existing outlier detection methods
- Clear and well-organized paper

Weaknesses:
- Mixed results for OOD detection
- The baselines used for comparison are questionable since they do not align with reports in the literature
- The setup is not self-supervised as there is explicit supervision by labeling pixels into the additional class
- The experimental setup is unclear, and the differences between the presented results and those in the literature are not clearly explained.

While the paper received a wide range of recommendations and the understanding of the problem varies among reviewers, there are major concerns regarding the validity of the experimental results. Despite the authors' efforts during the rebuttal, there are still concerns about the validity of the experiments, and it is unclear why the setups differ from the literature, leading to different results. Moreover, the strengths raised do not outweigh these concerns. The proposal is borderline, but due to the lack of stronger reasons to accept the paper, I lean towards rejecting it.

**Additional Comments On Reviewer Discussion:**

Reviewer wmG8 mentions that the results are mixed for OOD detection and raises concerns regarding the lack of clear training setups (self-supervised vs. supervised) given the labels. Moreover, the experiments are missing current methods for comparison. The authors presented several clarifications during the rebuttal, and the reviewer raised the score to a weak accept based on them.

Reviewer R2z4 mentions that the motivation of the proposal is clear. However, the reviewer complains that the experiment results do not align with the original publications, raising questions about the validity of the experiments. The authors responded to the reviewer's concerns and explained the misconceptions. The reviewer offered to review the scores after the rebuttal phase.

Reviewer Wr5a mentioned that the definition of the task is unclear. It is unclear what the rationale of the reviewer was for obtaining a definition of the task by asking ChatGPT about a figure of the paper. The reviewer also raises concerns about limited technical contributions since the proposal mainly combines existing methodologies. The authors provided the reviewer with additional details and explanations. After the rebuttal, the reviewer maintains that the tasks require more than segmentation masks.

During the post-rebuttal discussion, Reviewer Wr5a briefly replied that the task is still unclear and did not comment on whether the task as defined by the authors is sound. The reviewer maintains that the paper is solved by large multimodal language models. Reviewer R2z4 also commented back and maintains a conservative weak reject, given the inconsistent results and lack of clarification from the authors.

Given the lack of understanding from Reviewer Wr5a, I would like to weigh that recommendation the least. Similarly, given the lack of engagement from Reviewer wmG8, it is hard to know if the strengths outweigh the raised problems by Reviewer R2z4. Overall, the paper is borderline, and the problems with the experiments make me lean towards rejecting the paper.

---

### Decision · Program_Chairs · 2025-01-22

Reject